# The Effect of Dietary Carbohydrate and Fat Manipulation on the Metabolome and Markers of Glucose and Insulin Metabolism: A Randomised Parallel Trial

**DOI:** 10.3390/nu14183691

**Published:** 2022-09-07

**Authors:** Deaglan McCullough, Tanja Harrison, Lynne M. Boddy, Kevin J. Enright, Farzad Amirabdollahian, Michael A. Schmidt, Katrina Doenges, Kevin Quinn, Nichole Reisdorph, Mohsen Mazidi, Katie E. Lane, Claire E. Stewart, Ian G. Davies

**Affiliations:** 1Carnegie School of Sport, Leeds Beckett University, Leeds LS6 3QS, UK; 2Research Institute of Sport and Exercise Science, Liverpool John Moores University, Liverpool L3 3AF, UK; 3Department of Clinical Sciences and Nutrition, University of Chester, Chester CH1 4BJ, UK; 4School of Public Health Studies, University of Wolverhampton, Wolverhampton WV1 1LY, UK; 5Advanced Pattern Analysis and Countermeasures Group, Boulder, CO 80302, USA; 6Sovaris Aerospace, Boulder, CO 80302, USA; 7Skaggs School of Pharmacy and Pharmaceutical Sciences, University of Colorado, Aurora, CO 80045, USA; 8Clinical Trial Service Unit and Epidemiological Studies Unit (CTSU), Nuffield Department of Population Health, University of Oxford, Oxford OX3 7LF, UK; 9Department of Twin Research & Genetic Epidemiology, South Wing St Thomas’, King’s College London, London SE1 7EH, UK

**Keywords:** insulin resistance, cardiometabolic risk, low carbohydrate, high-fat diet, carbohydrate, fat, metabolomics

## Abstract

High carbohydrate, lower fat (HCLF) diets are recommended to reduce cardiometabolic disease (CMD) but low carbohydrate high fat (LCHF) diets can be just as effective. The effect of LCHF on novel insulin resistance biomarkers and the metabolome has not been fully explored. The aim of this study was to investigate the impact of an *ad libitum* 8-week LCHF diet compared with a HCLF diet on CMD markers, the metabolome, and insulin resistance markers. *n* = 16 adults were randomly assigned to either LCHF (*n* = 8, <50 g CHO p/day) or HCLF diet (*n* = 8) for 8 weeks. At weeks 0, 4 and 8, participants provided fasted blood samples, measures of body composition, blood pressure and dietary intake. Samples were analysed for markers of cardiometabolic disease and underwent non-targeted metabolomic profiling. Both a LCHF and HCLF diet significantly (*p* < 0.01) improved fasting insulin, HOMA IR, rQUICKI and leptin/adiponectin ratio (*p* < 0.05) levels. Metabolomic profiling detected 3489 metabolites with 78 metabolites being differentially regulated, for example, an upregulation in lipid metabolites following the LCHF diet may indicate an increase in lipid transport and oxidation, improving insulin sensitivity. In conclusion, both diets may reduce type 2 diabetes risk albeit, a LCHF diet may enhance insulin sensitivity by increasing lipid oxidation.

## 1. Introduction

Insulin resistance is a complex metabolic disorder that increases the risk of type 2 diabetes (T2D) and is integral to cardiometabolic disease [1]. To combat cardiometabolic disease, Public Health England, the U.S. Department of Health and Human Services and the World Health Organization recommend a diet relatively high in carbohydrates (although low in sugar) and low to moderate in fat (particularly saturated fatty acids (SFA)) [2,3,4]. However, despite an increase in SFA intake, several studies have shown LCHF (<26% of energy intake) diets to be at least as effective as HCLF diets at improving metabolic risk factors, including plasma glucose and insulin concentrations [5,6,7,8,9]. Furthermore, a very LCHF diet (<10% of energy intake) showed greater improvements on fasting glucose levels compared to a HCLF diet independent of weight loss [10]. Although current evidence is equivocal, as both short-term LCHF and HCLF diets that induce weight loss have also demonstrated mixed responses to glucose/insulin metabolism [11,12]. Moreover, meta-analyses of long-term (>6 months) randomised control trials of healthy and dysglycaemic participants (fasting glucose levels > 5.6 mmol/L) reported no difference between LCHF or HCLF diets in improving fasting insulin or glucose levels [6,8].

Biomarkers such as fibroblast growth factor 21 (FGF21), adiponectin, C-reactive protein (CRP), leptin and plasminogen activated inhibitor-1 (PAI-1) are associated with cardiometabolic disease and insulin sensitivity [13,14,15,16,17]. For example, FGF21 and leptin are elevated in individuals with poor metabolic health, whereas adiponectin is reduced [13,18,19]. However, research is lacking on the impact of LCHF diets on these novel markers. In addition to such biomarkers, metabolomics is a powerful diagnostic and discovery tool that defines discrete chemical signatures of metabolites that are unique to lifestyle and disease states [20]. Distinct metabolites, consisting of amino acids, lipids, sugars and energy substrates are associated with glucose/insulin metabolism and diet and can predict future incidence of insulin resistance [21,22]. For instance, compared with a low-fat/high glycaemic index (GI) diet, a low GI diet in participants with obesity reduced branched-chain amino acids (leucine and tyrosine) and lipid species (lysophosphatidylcholines and phosphatidylcholine), which were negatively correlated with insulin resistance [23]. Consequently, metabolomic profiling of dietary interventions allows for a greater mechanistic understanding of the impact of diet on health status [24,25].

Therefore, this study aims to investigate the impact of an *ad libitum* 8-week LCHF diet compared with a HCLF diet (current UK guidelines) on cardiometabolic risk factors, the plasma metabolome, and markers of glucose and insulin metabolism in adults with a slightly elevated cardiometabolic risk. We hypothesise that the LCHF diet will show greater improvements in glucose and insulin metabolism and these improvements will be associated with a favourable level of cardiometabolic risk markers and metabolomic signature.

## 2. Materials and Methods

All procedures followed the CONSORT guidelines for reporting of randomised trials (Appendix A) [26].

### 2.1. Participant Recruitment and Screening

Participants were recruited from July 2017 to December 2018 through posters, emails, word of mouth and presentations. Participants were included if they were aged 19–64 years with a BMI of 18.5–29.9 kg/m^2^ and excluded if they were a smoker, vegan/vegetarian, took dietary supplements, had any known food allergies or intolerances, consumed alcohol above the weekly UK recommendations, were pregnant, suffered from an eating disorder, suffered from current or previous renal impairment, had a history of cardiometabolic diseases, or took lipid, blood pressure or blood glucose-lowering medication. Eligible participants were invited to arrive fasted for 12 h for an in-person screening and after providing informed consent, a fasted finger prick blood test (Cholestech LDX^®^, Alere Ltd., Stockport, UK) was collected and blood pressure and waist measurements were taken, to assess participants cardiometabolic risk (Appendix A). Cardiometabolic risk was calculated based on a points system modified from the RISCK (Reading, Imperial, Surrey, Cambridge, and Kings) trial [27]. A score between 4 and 11 indicated a small-elevated cardiometabolic risk (Appendix A).

All procedures were approved by the Liverpool John Moores University research ethics committee (REC number: 16/ELS/029) and all participants provided written informed consent. This study is registered with clinical trials, ‘CALIBER (Carbohydrates, Lipids and Biomarkers of Traditional and Emerging Cardiometabolic Risk Factors)’ (REF: NCT03257085).

### 2.2. Study Protocol and Diet

After screening, in a parallel randomised design, participants were randomly assigned to either a HCLF (*n* = 8), [2] or a LCHF diet (*n* = 8) for 8 weeks using a computerised random allocation sequence, concealed in envelopes. Participants were given a personalised meal plan, recipes, and a detailed guide to follow each diet by a registered nutritionist (Appendix A). Participants in the HCLF group were required to consume a diet composed of 50% carbohydrate, 15% protein, and at most 35% fat per day (based on the UK Eatwell Guide) [2]. This equated to approximately 333 g and 267 g of carbohydrate, 55 g and 45 g of protein and at most 97 g and 78 g of fat per average male and female energy requirements of 2500 kcal and 2000 kcal per day respectively [28]. The diet also was required to be high in fibre and low in free sugars. The LCHF group were instructed to consume a diet consisting of ≤50 g of carbohydrate per day to induce ketosis [29], and increase the amount of fat consumed while eating similar amounts of protein to the HCLF group. Participants were also provided with guidance on how to increase fibre intake while also keeping carbohydrate low. Examples of foods consumed by the LCHF participants included unlimited amounts of beef, poultry, fish, eggs, oils, and heavy cream; moderate amounts of hard cheeses, low carbohydrate vegetables and salad dressings; and small amounts of nuts, nut butters and seeds. Participants restricted the following food: fruit and fruit juices, dairy products (except for heavy cream and hard cheese), breads, grains, pasta, cereal, high carbohydrate vegetables, and desserts. Subjects were instructed to avoid all low-carbohydrate breads and cereal products. To maintain healthy micronutrient levels all participants in the LCHF group were provided and instructed to consume a daily multivitamin (Centrum Women, Centrum, Pfizer Consumer Healthcare Ltd., Brentford, Middx, UK). No specific instructions on caloric intake were given to participants in both groups, as they were required to eat *ad libitum* to satiety. Participants were also contacted at least every week by telephone or email to check if assistance was needed with recipes or meals to promote adherence.

Participants were required to attend LJMU laboratories in the morning after fasting for 12 h at 0, 4 and 8 weeks of the diet. On each testing day, anthropometrics, a venous blood sample, blood pressure and body composition were collected. During the week prior to each testing day participants’ physical activity was measured for 7 days by wearing an activity watch on their non-dominant arm (Actigraph GT9X, Pensacola FL USA). On the 2nd and final testing day participants completed a fibre intake questionnaire and both groups were requested to report any adverse events associated with the diet.

### 2.3. Biochemical Measurements

Whole blood was collected by trained phlebotomists from the antecubital fossa vein and centrifuged at 3000× *g* for 15 min and 4 °C to harvest plasma and serum, which was stored at −80 °C until analysis.

To determine the following plasma cytokines and hormones: ferritin, interleukin-6 (IL-6), insulin, Leptin, PAI-1, resistin, tumour necrosis factor-α (TNFα), adiponectin, CRP and cystatin C (CYSC), the Evidence Investigator (EI)^TM^ Biochip Array technology (Randox Laboratories Ltd., Antrim, UK) was used. Plasma glucose, non-esterified fatty acids (NEFA) and ketone D-3 hydroxybutyrate were assessed with the Daytona, an automated random-access clinical chemistry analyser (Randox Laboratories Ltd., Antrim, UK). FGF21 was detected by ELISA analysis (R & D Systems, Minneapolis, MN, USA). The homeostatic model of insulin resistance (HOMA IR) was calculated (fasting plasma insulin x fasting plasma glucose)/22.5) [30]. The revised quantitative insulin sensitivity check index (rQUICKI) was calculated by 1/log(fasting plasma glucose) + log(fasting plasma insulin) + log(fasting plasma NEFA) to measure insulin sensitivity [31].

### 2.4. Measurement of Blood Pressure, Body Composition and Dietary Intake

Blood pressure was monitored 3 times using a digital sphygmomanometer (Omron Healthcare Europe B.V.), systolic and diastolic measures were recorded. Body composition was measured using bioelectrical impedance (SECA mBCA 515, Hamburg, Germany). Waist measurements (SECA 201 measuring tape, Hamburg, Germany) were recorded 3 times to the nearest 0.1 cm at the approximate midpoint between the lower margin of the last palpable rib and the top of the iliac crest [32]. Overall compliance with the dietary recommendations over the course of the intervention was assessed via 4-day food diaries [33]. Participants were asked to record all foods and beverages consumed during 2 weekdays and 2 weekend days. Participants were provided with a manual on how to complete the food diary in addition to examples of serving sizes and meal descriptions and were encouraged to be as detailed as possible. All were completed on 3 occasions at 0, 4 and 8 weeks. Analysis of food diaries was completed with Dietplan 7 (Forestfield Software Ltd., Horsham, SXW, UK).

### 2.5. Measurement of Metabolites

Plasma samples (100 µL) were subjected to a modified methyl tert-butyl ether extraction with a protein crash [34,35]. The lipid fraction was reconstituted in 200 µL methanol and the aqueous fraction was reconstituted in 100 µL 5% acetonitrile in water. The lipid and aqueous fractions were run in positive ionization mode on the Agilent Technologies 6560 Ion Mobility Q-TOF-MS, in Q-TOF only mode (Santa Clara, CA, USA).

Lipid fraction analytical column Agilent Technologies Zorbax Rapid Resolution HD (RRHD) SB-C18, 1.8 µm, 2.1 × 100 mm was used with a gradient elution: 0–0.5 min 30–70% B, 0.5–7.42 min 70–100% B, 7.42–10.4 min 100% B, 10.4–10.5 min 100–30% B, 10.5–15.1 min 30% B and a flow rate of 0.7 mL/min. Mobile phase A was prepared using water with 0.1% formic acid. Mobile phase B was prepared using 60:36:4 isopropyl alcohol:acetonitrile:water with 0.1% formic acid.

Aqueous fraction analytical column Agilent Technologies Poroshell 120 HILIC-Z, 2.1 × 100 mm, 2.7 µm was used with a gradient elution of 0–10 min 100–70% B, 10–11 min 70–100% B, 11–16 min 100% B and a flow rate of 0.8 mL/min. Mobile phase A was prepared using 100% water with 20 mM ammonium formate. Mobile phase B was prepared using 90% acetonitrile with 10% water with 20 mM ammonium formate.

Spectral peaks were extracted using Mass Hunter Profinder version B.10 Service Pack 1 (Agilent Technologies), using a Recursive Workflow and area-to-height conversion. The final spectral data was imported into Mass Profiler Professional (MPP) version 15.1 (Agilent Technologies). Annotation was performed with ID Browser (Agilent Technologies), using an in-house Mass and Retention Time (MSRT) database, HMDB databases, Lipid Maps database, KEGG databases and a Metiln database.

### 2.6. Statistics

All normally distributed data are presented as mean ± SD whereas non-normally distributed data are presented as medians ± interquartile range (IQR). All data were explored for distribution using the Shapiro-Wilks test. Normally distributed data underwent a 2 × 3 mixed ANOVA with 2 between factors (LCHF vs. HCLF) and 3 within factors (Baseline vs. interim vs. endpoint) to investigate significant differences for main and interaction effects. If repeated measures data had a missing value, mixed effects analysis was used instead of ANOVA. To assess the influence of possible baseline differences, baseline BMI was also used as a covariate for ANCOVA analysis. Non-normally distributed data underwent Mann-Whitney U tests at each time point to investigate differences between groups, Friedman’s test for differences within groups throughout the diets and Dunn’s test for post-hoc analysis. Baseline differences were assessed using independent *t*-tests. The alpha level for significance was set at *p* < 0.05 and GraphPad Prism v9 (San Diego, CA, USA) and IBM SPSS v27 (Armonk, NY, USA) statistical software was used for statistical analysis. R studio 1.1.463 was used to analyse the raw accelerometer data using GGIR [36]. Random forest with a combination of unbiased variable selection framework and repeated double cross-validation was applied to detect a panel of metabolites representative of the LCHF diet relative to the HCLF diet at the end of the study (week 8). This method fits many classification trees to a data set, and then combines the predictions from all trees to present a final predictive model that ranks variables by their predictive power. For the evaluation of our models, we have used R^2^ and Q^2^ (an estimate of the predictive ability of the model calculated by cross-validation). Model performance was confirmed by permutation analysis (*n* = 1000).

## 3. Results

Out of the 58 people screened, 41 attended the clinical screening, and 22 were excluded leaving 19 participants randomly assigned to either diet (Figure 1). Two participants withdrew before and another after baseline measurements due to time constraints. A further participant withdrew from the study after the interim time point due to illness, most likely due to following the LCHF diet, therefore 15 participants completed the intervention but all 16 were included in the intention to treat analysis. Per protocol analysis was also carried out but this did not affect the results (Appendix A).

Participants consisted of 9 males and 7 females with *n* = 8 per dietary group. Baseline measurements are shown in Table 1. Although there were significant differences in body composition (body mass, *p* = 0.037; body mass index (BMI), *p* = 0.018; fat mass %, *p* = 0.010 and fat free mass (FFM) %, *p* = 0.010) between groups, leptin and leptin/adiponectin ratio (LAR) were the only biochemical markers significantly higher in the LCHF group (Leptin, 1.8-fold; LAR, 3.2-fold; both *p* = 0.007) compared to the HCLF group at baseline. Dietary intake analysis showed no differences in energy or macronutrient composition at baseline (Appendix A).

### 3.1. Cardiometabolic Risk Factors

Both the LCHF and HCLF diets exerted benefits on markers of insulin resistance and metabolic risk (Table 2). Both diets had no effect on fasting glucose levels (*p* = 0.844), but insulin concentrations (Figure 2a) significantly (*p* = 0.002) decreased comparably with both diets, from 67.21 ± 27.64 to 55.53 ± 12.77 to 41.13 ± 17.40 pmol/L in the LCHF group and from 60.26 ± 15.48 to 52.44 ± 8.42 to 44.02 ± 11.23 pmol/L in the HCLF group at week 0, 4 and 8 respectively. This resulted in a significant (*p* = 0.008) decrease in HOMA IR (Figure 2b) by 1.1 and 0.6 points within the LCHF and HCLF diets, respectively. Furthermore, rQUICKI (Figure 2c) significantly (*p* = 0.003) increased similarly with both diets, from 0.35 ± 0.03 to 0.34 ± 0.01 to 0.37 ± 0.04 in the LCHF group and from 0.35 ± 0.03 to 0.36 ± 0.02 to 0.38 ± 0.04 in the HCLF group at week 0, 4 and 8, respectively. No difference at any timepoint or interaction between groups was observed in fasting glucose, insulin, HOMA IR and QUICKI.

FGF21 showed a trend (*p* = 0.051) in overall change in participants following the LCHF diet after decreasing from a median (IQR) of 148.16 (203.49) to 99.4 (92.43) pg/mL at week 4 and increasing again to 167.38 (152.08) pg/mL at week 8, with post-hoc analysis reporting a trend (*p* = 0.069) between baseline and week 4, whereas no change was observed in HCLF. FGF21 showed no difference between groups at any time point. The leptin/adiponectin ratio (LAR) was significantly higher at baseline (*p* = 0.007) and week 4 (*p* = 0.010) in the LCHF group compared with the HCLF group. The changes observed in adiponectin and leptin with both diets led to a significant decrease in LAR with the LCHF diet (*p* = 0.001) from a median (IQR) of 1.70 (2.63) ng/ug at baseline to 0.96 (1.4) ng/ug at week 4 to 0.70 (1.37) ng/ug at week 8 and the HCLF diet (*p* = 0.029) from a median (IQR) of 0.37 ng/ug (0.41) at baseline to 0.23 (0.28) ng/ug at week 4 to 0.31 (0.7) ng/ug at week 8. Post-hoc analysis reported LAR levels to be significantly lower at week 4 (*p* = 0.049) and week 8 (*p* = 0.004) compared with baseline in the LCHF group; however, in the HCLF group only week 4 showed a decreasing trend (*p* = 0.053) compared to baseline. Ferritin showed a similar significant (*p* = 0.039) reduction following the LCHF diet (Baseline: 172.61 ± 168.17 ng/mL; week 4, 162.49 ± 159.66 ng/mL; week 8, 173.14 ± 162.87 ng/mL) and HCLF diet (Baseline, 155.38 ± 83.3 ng/mL; week 4, 128.14 ± 72.66 ng/mL; week 8, 141.38 ± 89.82 ng/mL). No difference was observed between groups, and post-hoc analysis reported only a significant (*p* = 0.039) decreased from baseline to week 4 in the HCLF diet alone. Plasma D-3 hydroxybutyrate, a marker of ketosis, significantly (*p* = 0.035) increased only within the LCHF group from a median (IQR) of 0.07 mmol/L (0.01) at baseline to 0.31 (0.06) mmol/L at week 4 to 0.19 (0.27) mmol/L at week 8. At week 4 plasma D-3 hydroxybutyrate was significantly (*p* = 0.027) higher than the HCLF and post hoc analysis showed significantly (*p* = 0.049) higher levels compared with baseline with the LCHF diet only at week 4. To account for the potential effects of the different baseline BMI levels in each group on cardiometabolic health markers, ANCOVA analysis using baseline BMI as a covariate did not change the primary results suggesting baseline BMI did not influence findings (Appendix A). However, such analysis could not be performed on non-parametric data, which may have been influenced by baseline BMI and therefore interpreted with caution. 

### 3.2. Diet, Body Composition and Physical Activity Measures

No change in dietary intake was reported in the HCLF group during the intervention; however, as intended, the percentage of energy derived from fat and carbohydrate significantly (*p* < 0.001) increased from 34% to 61% and decreased from 42% to 10% respectively in the LCHF group (Appendix A). Percentage protein of total energy intake also significantly increased from 17% to 24% in the LCHF group. This resulted in percentage carbohydrate (*p* < 0.001), including total sugars (*p* < 0.001) being significantly lower, and fats (*p* < 0.001) and protein (*p* < 0.001) being significantly higher in the LCHF intervention compared with the HCLF diet. Total energy intake was similar between groups and did not significantly change during the intervention.

Body composition improved in the LCHF group but remained unchanged within the HCLF group during the intervention (Appendix A). In the LCHF group, VAT showed a tendency (*p* = 0.052) of decreasing, with post-hoc analysis showing a significant (*p* = 0.049) decrease at endpoint vs. baseline only. No change in VAT was observed in the HCLF group and no difference between groups was reported at any time point. There was no difference at any timepoint in total physical activity or total moderate to vigorous intensity between groups or within groups (Appendix A).

### 3.3. Non-Targeted Metabolomic Profiling

Non-targeted metabolomic profiling detected 3489 plasma metabolites in total. Following permutation tests, there was a clear differentiation in the plasma metabolome between 78 metabolites consisting of 22 lipid species, 14 amino acids and 42 other molecules in the LCHF diet compared with the HCLF diet (Appendix A). The 10 most influential metabolites distinguishing the LCHF diet from the HCLF diet were annotated (Metabolomics Standard Initiative level 3 ID) [37] as follows: Hydroxybutyrylcarnitine, N(Pai)-Methyl-L-Histidine, acetyl-L-Carnitine, ursodeoxycholic acid 3-sulfate, linoleic acid, N-(1-Deoxy-1-fructosyl) histidine, palmitoleyl linolenate, succinylcholine, 1-Aminocyclohexanecarboxylic acid, 5’-Hydroxymethyl meloxicam, (Table 3).

### 3.4. Adverse Events

One participant dropped out of the study after 5 weeks due to developing insomnia most likely from following the LCHF diet. The participant reported having insomnia on occasion before the intervention and felt following the LCHF diet had a worsening effect on their sleep.

## 4. Discussion

We show that both a LCHF and HCLF diet result in similar improvements in markers of insulin resistance via contrasting mechanisms as shown by the differential regulation of discrete metabolites, which may help describe how dietary carbohydrate and fat manipulation affect cardiometabolic health.

In accordance with previous research [5,7,8,38] but in contrast with dietary guidelines [2,3,4], a LCHF diet (with an approximately 2-fold increase in SFA) demonstrated significant (*p* < 0.01) improvements in markers of IR such as fasting insulin levels, HOMA IR and rQUICKI signalling reduced T2D risk [30,31]. Interestingly, in the absence of weight loss or significant change in macronutrient composition, similar improvements were also observed in insulin resistance markers by the HCLF diet—indicating a personalised diet approach could be used to address the growing prevalence of T2D [39]. Poor lifestyle habits and elevated VAT increase circulating free fatty acids, inflammatory cytokines and reactive oxygen species—biochemical mechanisms that contribute to IR [1,40]. Our data suggest both LCHF and HCLF diets modulate molecules related to restoring insulin sensitivity.

To uncover these mechanisms non-targeted metabolomics allows for discrete molecular signatures to be detected in response to changes in metabolism [20]. We found 3489 plasma metabolites, of which 78 metabolites were observed to be differentially regulated following the LCHF diet vs. the HCLF diet. These metabolites consisted of 22 lipid species, 14 amino acids and 42 other molecules. Compared with the HCLF, the LCHF diet triggered an increase in the lipid species hydroxybutyrylcarnitine (127%), palmitoleyl linolenate (66%) and acetyl-L-carnitine (47%) and decreased linoleic acid (−43%) abundance. Hydroxybutyrylcarnitine (HB-carnitine) is a carnitine ester produced from a synthase CoA reaction of hydroxybutyrate [41,42]. We showed the LCHF diet resulted in an increase in the ketone body D3-hydoxybutyrate as a result of carbohydrate restriction, which increases fatty acid flux and may have led to elevated HB-carnitine levels [41]. Not only are acylcarnitines elevated in obesity and T2D, but HB-carnitine concentrations can also distinguish between normal glucose tolerance, prediabetes and T2D [22,43,44]. An increase in fatty acid flux leads to increased but incomplete β-oxidation of fatty acids, increasing fatty acid oxidation intermediates such as HB-carnitines leading to insulin resistance [1,22,42]. However, we showed markers of insulin resistance improved; therefore, elevated HB-carnitine levels are likely a reflection of increased ketone bodies that may prevent the accumulation and trapping of mitochondrial acyl CoAs, allowing the continuation of CoA-dependent metabolic processes to enhance lipid transport and oxidation (particularly important due to the high FFA flux) [41,45]. The increase in lipid metabolites palmitoleyl linolenate and acetyl-L-carnitine following the LCHF diet may also signal an increase in greater transport and complete fatty acid oxidation, to reduce overaccumulation of fatty acid intermediates thus improving insulin sensitivity [46]. For example, in individuals with insulin resistance, increased acetyl-L-carnitine concentrations resulted in improved glucose tolerance, metabolic flexibility and markers of insulin resistance [46,47].

In contrast, linoleic acid, showed a 49% decrease following the LCHF diet. The evidence regarding linoleic acid with obesity and insulin resistance is conflicting. While some observational and metabolomic studies show inverse associations between obesity and insulin resistance others show the converse [22,48,49,50,51]. Likewise, mechanistically, evidence is equivocal on whether linoleic acid promotes or protects against insulin resistance. On one hand, data shows linoleic acid may improve insulin sensitivity by activating the Akt and AMP-activated protein kinase signalling pathways [52]. However, corn oil (rich in linoleic acid) can induce insulin resistance, in mice, via a lack of muscle peroxisome proliferator-activated receptor activation, and linoleic acid can induce inflammation, via lipid-mediated mechanisms [53], potentially leading to insulin resistance [54]. Importantly, the ratio of *n*-6 to *n*-3 fatty acids may also be of significance [55]. The current ratio of *n*-6/*n*-3 in the western diet is between 20-50/1 a contrast to ancestral estimations of 1-2/1, showing an excess of *n*-6 fatty acids [53,56]. *n*-3 fatty acids demonstrate anti-inflammatory qualities but compete with *n*-6 fatty acids to exert their metabolic effects [55]. Therefore, a high *n*-6/*n*-3 ratio can lead to chronic low-grade inflammation via pro-inflammatory eicosanoids, promoting insulin resistance [53]. Indeed, a recent meta-analysis showed a low ratio of *n*-6/*n*-3 fatty acid reduced insulin levels and insulin resistance [57]. Therefore, the reduced linoleic acid concentrations may indicate an improved ratio of *n*-6 to *n*-3 fatty acids leading to enhanced anti-inflammatory action of *n*-3 fatty acids and reduced insulin resistance [55,58]. In addition to lipid metabolites, amino acids and derivatives show positive associations with obesity and T2D [21,59]. Conversely, in the current study, metabolites of histidine (N-(1-Deoxy-1-fructosyl)histidine and N(Pai)-Methyl-L-Histidine) metabolism increased in both diets. Therefore, this may reflect a change in other metabolic processes instead of insulin sensitivity such as myofibrillar proteolysis [60]. However, fat-free mass remained unaltered following both diets. Both diets, although more so with the HCLF diet, decreased bile acid metabolite concentrations which is in accordance with previous research that bile acids are positively associated with insulin resistance [61].

Pro-inflammatory cytokines such as CRP, interleukin-6 (IL-6) and tumour necrosis factor alpha (TNF-α) are derived from immune cells or adipocytes and are implicated in inhibiting the insulin signalling cascade [62] whilst also positively associated with insulin resistance [17,63,64]. Our study and others [65] show no significant change in these inflammatory markers indicating a lack of a mechanistic role in regulating insulin sensitivity. Adiponectin and leptin are adipokines that regulate insulin sensitivity [19,66,67,68] and show contrasting responses to weight loss and cardiometabolic disease [69]. Consequently, the LAR is a reliable and useful marker in predicting markers of cardiometabolic health [70,71]. The LAR has shown to be highly correlated with waist circumference, insulin, HOMA IR, and T2D [70]. The present study showed a significant improvement in the LAR with the LCHF (*p <* 0.001) and HCLF (*p =* 0.029) diets, suggesting an improvement in adipocyte function and insulin signalling. Although adiponectin and leptin are regularly measured in LCHF dietary intervention studies, there is currently a lack of use of the LAR, indicating a novel observation in the current study. In T2D patients following either a low-fat diet (*n* = 67), a Mediterranean diet supplemented with olive oil (*n* = 74) or mixed nuts (*n* = 50) for 1 year all showed similar improvements in LAR with reductions in waist circumference [72]. In obese participants (*n* = 10) following a very low-calorie diet, LAR significantly improved along with HOMA IR and body composition [73]. These and our results indicate that a diet that improves adipocyte function will improve insulin signalling.

FGF21 regulates metabolic energy homeostasis and animal studies show increases with starvation and ketogenesis [74]. In humans with obesity or cardiometabolic disease FGF21 is elevated, indicating potential FGF21 resistance, with no increase with ketogenesis [13,75]. Our study showed that FGF21 decreased (*p* = 0.05) only with the LCHF diet at midpoint but rebounded by the endpoint. The decrease in FGF21 levels in response to a ketogenic diet has been reported by Christodoulides et al. [76] in which 7 subjects (5 diabetic) reduced their FGF21 levels (*p* < 0.051) after 4 months. In the current study, the increase in FGF21 experienced by the participants at the endpoint may be attributed to a lack of adherence. Although the food diaries suggest adherence was maintained, measurement of D-3 hydroxybutyrate, an indicator of ketosis, was highest at 4 weeks (0.31 ± 0.06 mmol/L) and decreased by 8 weeks (0.19 ± 0.27 mmol/L). Yet, these increased ketone levels would only be considered small as they are below the recommended ketosis threshold of 0.5 mmol/L and therefore unlikely to have a significant metabolic effect [77]. Christodoulides et al. [76] reported higher levels of ketones (0.5 ± 0.5 mmol/L) throughout, which may have maintained the reduction in FGF21. We speculate we may have maintained our reduction in FGF21 with higher nutritional ketosis. However, despite this, we still found differences in VAT and other body mass measures. Indeed, conventional HCLF diets also reduce FGF21 when in tandem with decreased body fat [75], suggesting weight loss alone rather than diet composition contributes to reduced FGF21 levels.

Like all studies, the current study contains some limitations. The study was small with respect to sample size with no formal power calculation and should be considered as a pilot, that explored plasma metabolites with non-targeted metabolomics, and therefore interpreted with caution. Furthermore, there may have been sex differences that influence various metabolic markers, but we feel the sample size was too small to explore this. The study was conducted in non-obese, non-diabetic individuals and therefore, the results should not be generalised to T2D or obese populations. This study is also short in duration limiting extrapolation of findings to long-term health; however, the objective of this study was to investigate changes in metabolic health and associated mechanisms rather than long-term compliance and these changes were evident. Participants were randomly assigned to either group, yet unfortunately, this has led to a possible imbalance in baseline measures as the LCHF group was significantly (*p* < 0.05) heavier due to increased fat mass. This may explain some of the improvements within the LCHF diet as they may have had higher levels of metabolic disease at baseline. Nonetheless, all participants appeared to be metabolically similar at baseline and ANCOVA analysis did not alter the results. However, not all CMD markers could undergo ANCOVA analysis to rule out the effect of baseline differences. While the strengths of the study include the use of an unbiased non-targeted metabolomic approach to identify key metabolic signatures that can identify disease risk and help elucidate potential dietary-related mechanisms. However, samples were shipped on dry ice from the UK to Colorado USA for metabolomics, and while samples were still cold, they had thawed. This may have impacted on results.

In conclusion, following either a LCHF or HCLF diet may reduce the risk of developing T2D by reducing markers of insulin resistance. Although increased fatty acid intermediates are a source of insulin resistance, a LCHF diet may increase fatty acid transport and oxidation thus reducing the risk of insulin resistance. Further research is warranted on the potential long-term mechanisms of reducing cardiometabolic risk through manipulating dietary macronutrient intake.

## Figures and Tables

**Figure 1 nutrients-14-03691-f001:**
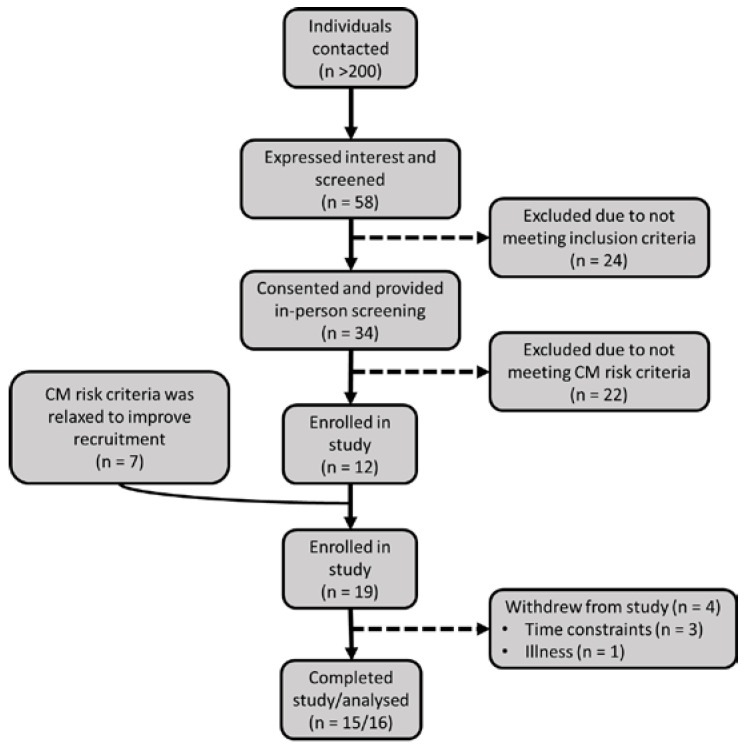
Study flow chart. CM: cardiometabolic.

**Figure 2 nutrients-14-03691-f002:**
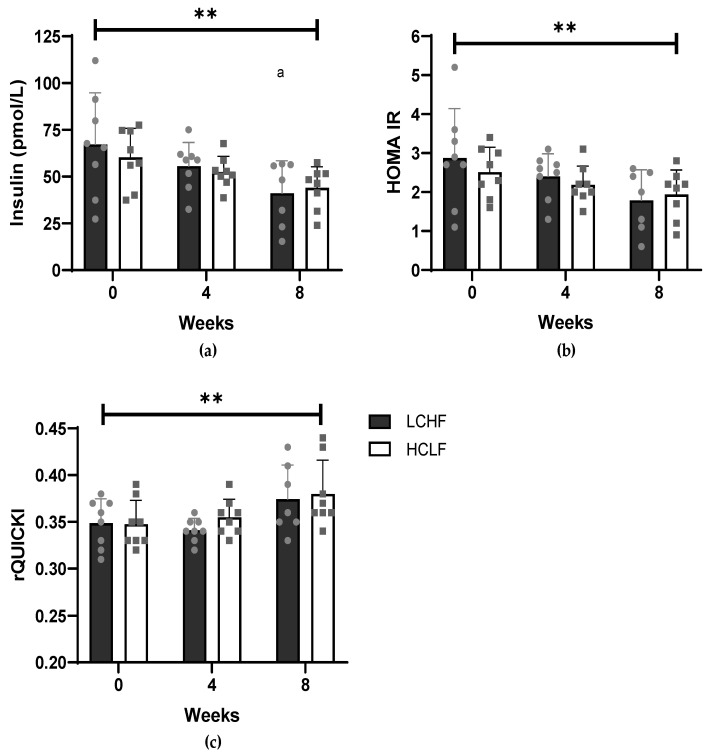
The effect of a LCHF (*n* = 8) and HCLF diet (*n* = 8) for 8 weeks on markers of insulin resistance. (**a**) The effect of a LCHF and HCLF diet on fasting insulin concentrations. (**b**) The effect of a LCHF and HCLF diet on HOMA IR. (**c**) The effect of a LCHF and HCLF diet on rQUICKI. The grey symbols represent individual responses of the LCHF diet (circle) and HCLF diet (square). HCLF, high carbohydrate low fat; HOMA IR, homeostatic model of insulin resistance, LCHF, low carbohydrate high fat; rQUICKI, revised Quantitative Insulin sensitivity Check Index. The straight line represents the effect of time from the 2 × 3 mixed ANOVA (** *p* < 0.01), ^a^
*p* < 0.05, denotes significantly different to week 0.

**Table 1 nutrients-14-03691-t001:** Baseline differences between the LCHF (*n* = 8) and HCLF (*n* = 8) groups.

	LCHF	HCLF	
	Baseline		(*p* Value)
Age M/F ratio	43.8 ± 10.4 4/4	44.6 ± 15.27 5/3	0.895 -
Height (cm)	176 ± 11.6	168 ± 11.02	-
Mass (kg)	89.53 ± 14.85	72.48 ± 14.72	0.037
BMI (kg/m^2^)	28.75 ± 2.15	24.56 ± 3.83	0.018
FM (kg)	32.08 ± 4.28	19.89 ± 7.46	0.001
FFM (kg)	57.39 ± 11.62	52.59 ± 11.63	0.423
FM (%) ^$^	35.65 (7.43)	25.5 (2.5)	0.010
FFM (%) ^$^	64.2 (7.40)	74.5 (2.50)	0.010
SkM (%)	32.05 ± 4.92	35.8 ± 4.20	0.071
SBP (mmHg)	130 ± 10	127 ± 12	0.673
DBP (mmHg)	83 ± 9	78 ± 8	0.373
VAT (I) ^$^	3.21 (3.43)	2.65 (1.35)	0.328
WC (cm) ^$^	101.2 (15.87)	93.75 (22.34)	0.054
Glucose (mmol/L) ^$^	5.47 (0.75)	5.72 (0.59)	0.520
Adiponectin (ug/mL) ^$^	2.11 (1.98)	2.69 (3.27)	0.442
CRP (ng/mL)	1.57 ± 0.83	1.25 ± 0.50	0.359
Ferritin (ng/mL)	172.61 ± 168.17	155.38 ± 83.3	0.799
IL-6 (pg/mL) ^$^	1.16 (1.14)	1.11 (1.61)	0.574
Insulin (pmol/L)	67.21 ± 27.64	60.26 ± 15.48	0.545
HOMA IR	2.9 ± 1.30	2.5 ± 0.60	0.482
rQUICKI	0.35 ± 0.03	0.35 ± 0.03	0.924
PAI-1 (ng/mL) ^$^	7.19 (10.04)	5.14 (8.02)	0.721
Resistin (ng/mL) ^$^	2.76 (2.41)	3.82 (2.57)	>0.999
TNFa (pg/mL)	6.41 ± 0.69	6.52 ± 1.81	0.868
D3-Hydroxybutarate (mmol/L)	0.07 ± 0.02	0.08 ± 0.05	0.478
Cystatin C (pg/mL)	0.35 ± 0.11	0.41 ± 0.13	0.470
Leptin (ng/mL) ^$^	3.69 (2.01)	1.31 (0.71)	0.007
LAR (ng/ug) ^$^	1.58 (2.63)	0.37 (0.41)	0.007
FGF21 (pg/mL)	147.9 ± 114.6	180.7 ± 138.5	0.614

Values are expressed as means ± SD. ^$^ denotes values expressed as medians (interquartile range) (non-parametric data). Significance set at *p* < 0.05. BMI, Body mass index; CRP, C-Reactive protein; DBP, Diastolic blood pressure; FGF21, Fibroblast growth factor 21; FM, Fat Mass; FFM, Fat-free mass; HOMA IR, homeostatic model of insulin resistance; IL-6, interleukin-6; Leptin/Adiponectin ratio; PAI-1, Plasminogen activated inhibitor-1; rQUICKI, revised Quantitative Insulin sensitivity Check Index; SkM, Skeletal muscle mass; TNFα, Tumour necrosis factor-α; VAT, Visceral adipose tissue.

**Table 2 nutrients-14-03691-t002:** Changes in anthropometric variables and biochemical markers of metabolic health following the LCHF and HCLF diets for 8 weeks.

	LCHF			HCLF					
Measure	Week 0	Week 4	Week 8	Week 0	Week 4	Week 8	TIME	T X G	Group
SBP (mmHg)	130 ± 10	120 ± 10 ^b^	123 ± 9 ^b^	127 ± 12	128 ± 12	129 ± 14	0.040	0.010	0.431
DBP (mmHg)	83 ± 9	77 ± 9	75 ± 9 ^bb^	78 ± 8	79 ± 8	78 ± 9	0.009	0.009	0.870
Glucose (mmol/L)	5.69 ± 0.41	5.78 ± 0.47	5.68 ± 0.47	5.79 ± 0.35	5.60 ± 0.54	5.80 ± 0.53	0.844	0.159	0.925
Cystatin C (ug/mL)	0.36 ± 0.11	0.31 ± 0.06	0.36 ± 0.09	0.41 ± 0.13	0.43 ± 0.17	0.43 ± 0.15	0.620	0.201	0.183
NEFA (mmol/)	0.79 ± 0.34	0.96 ± 0.30	0.80 ± 0.24	0.76 ± 0.25	0.79 ± 0.22	0.62 ± 0.17	0.078	0.587	0.309
Ferritin (ng/mL)	172.61 ± 168.17	162.49 ± 159.66	173.14 ± 162.87	155.38 ± 83.30	128.14 ± 72.66	141.38 ± 89.82	0.039	0.290	0.718
Non-parametric testing							LCHF (Time)	HCLF (Time)	
WC (cm)	101 (15.8)	99.1 (17.2)	98.4 (16.9) ^b^	93.8 (22.4)	93.1 (21.9)	92.0 (22.1)	0.052	0.346	
VAT (l)	4.23 (3.09)	3.83 (3.09)	3.19 (3.06) ^b^	2.65 (1.98)	2.46 (1.90)	2.46 (1.84)	0.052	0.544	
Adiponectin (ug/mL)	2.69 (3.27)	3.33 (1.91)	3.19 (3.33)	2.01 (2.09)	2.12 (1.51)	2.12 (1.85)	0.964	0.079	
CRP (ug/mL)	1.10 (1.62)	1.69 (1.66) **	1.24 (1.29)	1.06 (0.79)	0.88 (0.51)	0.98 (0.86)	0.305	0.236	
IL-6 (pg/mL)	1.30 (1.37)	1.23 (1.27)	1.54 (0.38)	1.11 (1.61)	0.83 (0.92)	1.04 (1.99)	0.964	0.794	
TNFα (pg/mL)	6.74 (1.28)	5.83 (1.93)	5.97 (1.74)	6.50 (3.19)	6.23 (3.21)	6.15 (3.45)	0.620	0.531	
Leptin (ng/mL)	3.98 (1.86) **	1.98 (1.84) ^b^ **	1.20 (2.42) ^bb^	1.31 (0.71)	0.95 (0.54)	1.07 (0.86)	0.001	0.285	
LAR (ng/ug)	1.70 (2.63) **	0.96 (1.4) ^b^ **	0.70 (1.37) ^bb^	0.37 (0.41)	0.23 (0.28)	0.31 (0.7)	0.001	0.029	
Resistin (ng/mL)	2.75 (1.15)	2.76 (1.47)	2.60 (1.51)	3.82 (2.57)	2.93 (1.78)	2.97 (1.70)	0.964	0.150	
PAI-1 (ng/mL)	7.30 (9.61)	6.53 (4.45)	6.32 (5.20)	5.14 (8.02)	4.11 (5.37)	4.47 (9.99)	0.112	0.794	
FGF21 (pg/mL)	148.16 (203.49)	99.4 (92.43)	167.38 (152.08)	138 (243.56)	136.4 (233.41)	201.3 (406.55)	0.051	0.935	
D-3-Hydroxybutyrate (mmol/L)	0.07 (0.01)	0.31 (0.06) ^b^ *	0.19 (0.27)	0.09 (0.05)	0.13 (0.11)	0.08 (0.03)	0.035	0.580	

Values are expressed as means ± SD of *n* = 8 LCHF & *n* = 8 HCLF. Non-parametric testing values are expressed as median ((interquartile range) of *n* = 7 LCHF & *n* = 8 HCLF. ^b^
*p* < 0.05, ^bb^
*p* < 0.01, denotes significantly different to baseline, * *p* < 0.05, ** *p* < 0.01 denotes significantly different between groups at that timepoint. CRP, C-Reactive protein; CYSC, Cystatin-C; DBP, Diastolic blood pressure; HOMA IR, homeostatic model of insulin resistance; IL-6, interileukin-6; LAR, Leptin/Adiponectin ratio; NEFA, non-esterified fatty acids; PAI-1, Plasminogen activated inhibitor-1; SBP, systolic blood pressure; TNFα, Tumour necrosis factor-α; VAT, Visceral adipose tissue.

**Table 3 nutrients-14-03691-t003:** The top 10 differential responses of metabolites following a LCHF and HCLF diet after 8 weeks.

		Fold Change (%)	
Metabolite	Description	LCHF	HCLF
Hydroxybutyrlcarnitine	Lipid metabolism	121.83	−5.03
1-Aminocyclohexanecarboxylic acid	L-α-amino acid metabolism	−86.27	15.44
N-(1-Deoxy-1-fructosyl)histidine	Histidine metabolism	73.44	3.28
Palmitoleyl linolenate	Lipid Metabolism	52.13	−14.21
Succinylcholine	Acyl choline metabolism	−15.12	43.1
Acetyl-L-Carnitine	Lipid Metabolism	47.95	1.03
Ursodeoxycholic acid 3-sulfate	Bile acid metabolism	−31.94	−74.87
Linoleic acid	Lipid metabolism	−49.42	−6.62
5’-Hydroxymethyl meloxicam	Purine ribonucleoside metabolism	6.56	44.66
N(Pai)-Methyl-L-Histidine	Histidine metabolism	49.25	86.39

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
