# Peer review of "The Effect of Dietary Carbohydrate and Fat Manipulation on the Metabolome and Markers of Glucose and Insulin Metabolism: A Randomised Parallel Trial"

_nutrients, 2022, doi:10.3390/nu14183691_

Round 1

Reviewer 1 Report

The manuscript under review seeks to address the metabolic consequences of HCLF vs. LCHF diets in humans with no evident cardiometabolic diseases such as T2D. The evidence in this study adds some new insights to the longstanding debate over the efficacy of dietary interventions that manipulate macronutrient content in efforts to reduce cardiometabolic risk factors and cause weight loss. The study, even with acknowledged limitations, succeeds in providing new evidence showing associations between parameters such as leptin : adiponectin ratio and improved metabolic health concomitant with VAT reductions. Specific comments are below:

- The authors note that ketone levels were not robustly increased in the HCLF group. Although significant compared to baseline in the HCLF group, at no time during the study were ketones elevated to the range classically defined as nutritional ketosis (>0.5 mmol). This does limit the interpretation of the study in some aspects. If ketones, and therefore compliance, had been robustly increased it is fair to speculate that the differences between groups could have been more pronounced. However, on the other hand, it is notable that even without a potent induction of ketosis there were improvements in several biochemical and anthropometric measures (VAT loss for example). 

- The discussion of linoleic acid levels is confusing to this reviewer (lines 350-358).  The authors note that linoleic acid is inversely associated with obesity and insulin resistance in line 350 and yet go on to report that linoleic acid was reduced by 49% in the LCHF group, which is associated with desirable alterations in several outcomes of the study demonstrated in the LCHF group. There is one reference (50) provided asserting that linoleic acid exerts antidiabetic effects. Omega-6 fatty acid intake far in excess of omega-3 intake is positively associated with obesity and insulin resistance. There is a formidable body of evidence supporting the notion that high consumption of vegetable oils (corn and safflower, for example) abundant in linoleic acid is a risk factor for metabolic disease. The authors note that the ratio of n-6 to n-3 fatty acids could be an underlying factor that could account for the positive effects observed in the LCHF group and that the ratio of these fatty acids is an important variable. The discussion could be improved by adding more nuance and references to better explain that linoleic acid intake cannot simply be regarded as a benign player in nutrition. The discussion should take into account the importance of n-6 : n-3 ratio and provide more references and discussion pertaining to the role of linoleic acid intake in the etiology of insulin resistance. 

- The authors are careful to note the limitations of the study. Particularly noteworthy is the careful attention given to the failure of the HCLF diet to induce nutritional ketosis, thereby raising the issue of noncompliance with the dietary protocol. Unfortunately, in human trials conducted outside of a metabolic ward setting, noncompliance with dietary protocols is an inherent problem and it is a strength of the study that the authors measured blood ketone levels and provided a proper discussion of this limitation.

Reviewer 2 Report

The paper by McCullough and coll. reports the results of a comparison between HCLF and LCHF in terms of metabolic markers and of non-targeted metabolomics in a cohort of 16 adults. Both diets were effective in improving fasting insulin, HOMA IR, rQUICKI and leptin/adiponectin ratio. Metabolomic analysis identified 78 metabolites with different regulation in the 2 treatment groups.. LCHF specifically resulted in upregulation of lipid metabolites, indicating increased lipid transport and oxidation.

The paper is clearly written and understandable. The Introduction is sufficiently explanatory about the topic of the study.The M&M are well detailed. The Results are clearly explained. Particularly interesting appears the non-targeted metabolomic study applied to this field.

The Discussion is complete. In the section about LAR, in addition to ref. (64), the Authors may also add the following reference, reporting the correlation of LAR with IMT, another relevant marker of vascular damage:

Giuseppe Danilo Norata, Sara Raselli, Liliana Grigore, Katia Garlaschelli, Elena Dozio, Paolo Magni, Alberico L Catapano. Leptin:adiponectin ratio is an independent predictor of intima media thickness of the common carotid artery. Stroke. 2007 Oct;38(10):2844-6. doi: 10.1161/STROKEAHA.107.485540. PMID: 17823381

Specific comments

I do not see reported any power calculation: is the cohort size sufficient? Please explain.

The sex composition of each group does not seem to be reported. It is obvious that a different M:F ratio may affect the cumulative results in each group, especially when anthropometric parameters and body composition are concerned, but also leptin and other sex-related parameters. Please explain.

LCHF and HCLF differed by BMI, body mass and composition. It is unclear whether this resulted by the randomization. In any case, the two cohorts are rather different at baseline at least. In particular, HCLF had a mean BMI of 24.5 and so maybe they were around the upper level of normal weight/metabolic conditions. Please, explain how we should be able to compare the 2 cohorts.

Figure 2. The horizontal bar is unclear. Which are the 2 values that are compared? It seems that in the 3 panels the first column is compared with the very last.
